# HeLa Cervical Cancer Cells Are Maintained by Nephronophthisis 3-Associated Primary Cilium Formation via ROS-Induced ERK and HIF-1α Activation under Serum-Deprived Normoxic Condition

**DOI:** 10.3390/ijms232314500

**Published:** 2022-11-22

**Authors:** Jae-Wook Lee, Jun-Yeong Cho, Pham Xuan Thuy, Eun-Yi Moon

**Affiliations:** Department of Integrative Bioscience and Biotechnology, Sejong University, Seoul 05006, Republic of Korea

**Keywords:** nephronophthisis 3, primary cilium, ROS, ERK, HIF-1α

## Abstract

The primary cilium (PC) is a microtubule-based antenna-like organelle projecting from the surface of the cell membrane. We previously reported that PC formation could be regulated by nephronophthisis 3 (NPHP3) expression followed by its interaction with thymosin β4. Here, we investigated whether cancer cell viability is regulated by NPHP3-mediated PC formation. The total and viable cell number were reduced by incubating cells under serum deprivation (SD) without fetal bovine serum (-FBS). PC frequency was increased by SD which enhanced NPHP3 expression and hypoxia inducible factor (HIF)-1α. The role of HIF-1α on NPHP3 expression and PC formation was confirmed by the binding of HIF-1α to the NPHP3 promoter and siRNA-based inhibition of HIF-1α (siHIF-1α), respectively. HIF-1α-stabilizing dimethyloxallyl glycine (DMOG) and hypoxic conditions increased NPHP3 expression and PC formation. In addition, as SD elevated the reactive oxygen species (ROS), PC frequency and NPHP3 expression were inhibited by a treatment with N-acetylcysteine (NAC), a ROS scavenger. PC formation was increased by H_2_O_2_ treatment, which was inhibited by siHIF-1α. The inhibition of ERK with P98059 decreased the frequency of PC formation and NPHP3 expression. Cell viability was reduced by a treatment with ciliobrevin A (CilioA) to inhibit PC formation, which was re-affirmed by using PC-deficient IFT88^−/−^ cells. Taken together, the results imply that PC formation in cancer cells could be controlled by NPHP3 expression through ROS-induced HIF-1α and ERK activation under SD conditions. It suggests that cancer cell viability under SD conditions could be maintained by NPHP3 expression to regulate PC formation.

## 1. Introduction

Although the serum including growth factor is necessary for normal cell growth, cancer cells grow well irrespective of serum. Reactive oxygen species (ROS) were thought to be produced as byproducts of cellular respiration under serum deprivation (SD) [1] or nutrient deprivations [2]. Oxidative stress is defined as a disturbance in the balance between the production of ROS and antioxidant defenses [3]. Superoxide anion (O_2_^−^), hydrogen peroxide (H_2_O_2_) and the hydroxyl radical (HO•) are mainly involved in ROS and detoxified by ROS-regulating antioxidant enzymes such as superoxide dismutase (SOD), catalase, heme oxygenase-1 (HO-1) and reduced glutathione (GSH) [4]. Nuclear factor erythroid-derived 2-like 2 (Nrf2) is a master transcription factor to regulate gene expression of antioxidant enzymes [5,6]. SD-induced ROS provided an early oxidative stress environment and induced an apoptotic response in various cell types [7,8]. SD-induced ROS also allowed cancer cells to confer an adaptability to increased oxidative stress [9]. Various cellular events including primary cilium (PC) formation are also altered in SD condition [10].

The PC biogenesis is induced by incubation with a SD medium in many types of cultured cells [11,12,13,14]. The PC is a microtubule-based non-motile signaling organelle that grows in a specific region of the plasma membrane and senses changes in the enrichment of nutrients [15]. The PC is assembled in the G0/G1 phase, disassembled in the S phase and disappears in G2/M phase [16]. SD-induced cell cycle arrest at the G0/G1 phase is an appropriate condition to induce PC formation in vitro. However, the role of PC formation in cancer is still to be debated and the underlying mechanisms of PC formation to control cancer cell viability under SD conditions are not well-understood.

NPHP3 is one of the ciliary proteins localized in the basal body and centrioles of primary cilia. Mutations in NPHP3 are responsible for adolescent nephronophthisis (NPHP), which is an autosomal recessive poly cystic kidney disorder and the most frequent genetic disease of renal failure in children and young adults [17,18,19]. NPHP is considered to be one of the ciliopathies caused by ciliary dysfunction [20]. A homomorphic mutation of the NPHP3 allele was shown to be the result of a defect in the primary cilia length control in epithelial mouse kidney cells [21]. The knockdown of zebrafish ortholog NPHP3 with morpholino oligo reduces the frequency and the length of primary cilia in Kupffer’s vesicle [22]. However, no information regarding the role of NPHP3-associated PC formation to cancer cell viability has been reported.

In the adaptation of cells to SD-induced oxidative stress, hypoxia-inducible factor (HIF)-1α stabilization and HIF-1 activation inhibit apoptosis in cancer cells [23]. HIF-1 is composed of two subunits, HIF-1α and HIF-1β to control many hypoxia-inducible genes [24]. HIF-1 activity in tumors depends on the availability of the HIF-1α subunit. HIF-1α is a cytosolic protein and it is rapidly degraded by the ubiquitin-proteasome pathway in normoxia [25]. The degradation of HIF-1α is inhibited in hypoxia [26]. Hypoxia-induced mitochondrial ROS have been shown to be necessary for the stabilization of HIF-1 in hypoxic cells [27,28,29,30]. The nuclear translocation of the HIF-1α protein is enhanced under normoxic condition [31]. HIF-1α has also been shown to increase renal PC length following a renal injury [32]. Oxygen tension significantly affects the length of cilia in primary BMSCs with the re-localization of the SMO and GLI2 proteins to cilia [33]. Under normoxic conditions, SD resulted in a significant increase in cilia length [33]. It is required to define SD-associated molecules on PC formation and their cellular functions. For instance, thymosin beta-4 interacting with NPHP3 regulates tumor growth as well as PC formation [34,35]. However, no information regarding the role of HIF-1α on NPHP3-associated PC formation and cancer cell survival has been reported.

In this study, we investigated whether cancer cell viability was regulated by NPHP3 expression-associated primary cilium formation via ROS-induced HIF-1α under SD conditions in HeLa human cervical cancer cells. Our data demonstrated that PC formation regulated cancer cell viability under SD condition via NPHP3 expression by ROS-induced ERK activation and HIF-1α stabilization. 

## 2. Results

### 2.1. Serum-Deficiency Contributes to PC Formation and the Expression of NPHP3 and HIF-1α

A serum deficiency (SD) has induced apoptosis in many types of cells [36]. Nephronophthisis 3 (NPHP3) has interacted with thymosin beta-4 to regulate PC formation and tumor growth [34,35]. Cancer cell death has been inhibited by HIF-1α stabilization and activation in the adaptation of cells to SD-induced oxidative stress [23]. The nuclear translocation of the HIF-1α protein is enhanced under normoxic conditions [31]. To investigate the relationship between NPHP3 expression and HIF-1α under SD condition, we examined the SD effect on cell viability and the expression of PC, NPHP3 and HIF-1α. When HeLa cells were incubated in SD media for 36 h, the cell viability (Figure 1A) and total cell number (Figure 1B) reduced. PC was visualized by the observation and the merging of the separate images of each channel (Appendix A). Our experimental condition was verified by which PC formation was positively observed by the treatment with sodium butyrate for 48 h (Appendix A). Then, PC frequency in the HeLa cells was detected by counting over 500 cells in response to the incubation in SD medium (Figure 1C,D). The NPHP3 transcriptional increase was measured by its transcription (Figure 1E) and NPHP3 promoter activity (Figure 1F) in response to incubation in the SD medium. HIF-1α protein was also stabilized in response to the incubation in SD medium (Figure 1G top). In addition, the level of Nrf2, a positive marker protein for ROS production and the activation of antioxidant expression [5,6], was increased under SD condition (Figure 1G bottom). While the expression of SOD1 and catalase was increased ≈1.9 and ≈1.8 times, respectively, SOD2 and HO-1 expression was significantly decreased to ≈0.6 times (Figure 1H). The data suggested that SD-induced Nrf2 and HIF-1α could be related to NPHP3 expression leading to the regulation of cancer cell death. Here, the study will be focused on the effect of HIF-1α under normoxic SD condition.

### 2.2. HIF-1α Regulates NPHP3 Expression and PC Formation

We examined whether HIF-1α could regulate NPHP3 expression and PC formation. SD-induced HIF-1α interacted with the NPHP3 promoter as judged by a CHIP assay (Figure 2A). As HIF-1α expression was interfered with the first (Figure 2B top) or the second (Figure 2B bottom) siHIF-1α, the second siHIF-1α is more effective than the first siHIF-1α. The second siHIF-1α reduced the amount of HIF-1α protein (Figure 2C) and PC formation (Figure 2D). HIF-1α was stabilized ≈4.1 times by a treatment with DMOG, an inhibitor of proline hydroxylase (Figure 2E), which was inhibited ≈40% by the interference of HIF-1α with the second siHIF-1α (Figure 2F,G). The NPHP3 transcriptional increase with DMOG treatment was measured by its transcription (Figure 2H). DMOG-induced HIF-1α interacted with the NPHP3 promoter, as judged by a CHIP assay (Figure 2I). The number of PC-positive cells by DMOG treatment increased ≈1.7 times compared with the control (Figure 2J). Then, we confirmed the effect of HIF-1α on NPHP3 expression and PC formation using hypoxic condition as a well-known HIF-1α stabilizer. HIF-1α was stabilized with the incubation of cells under hypoxic conditions (Figure 3A). The NPHP3 protein (Figure 3B top), its transcription (Figure 3B bottom) and promoter activity (Figure 3C) also increased in response to incubation under hypoxic conditions. The relevance of HIF-1α to NPHP3 expression and PC formation was re-affirmed with the overexpression of GFP-labelled HIF-1α (Figure 3D). NPHP3 protein (Figure 3E top) and its transcription (Figure 3E bottom) increased, respectively, ≈1.4 and ≈2.1 times with the overexpression of GFP-labelled HIF-1α. The number of PC-positive cells by the HIF-1α-GFP overexpression increased ≈1.4 times compared with those in pEGFP-C1-transfected control group under normoxic conditions (Figure 3F). NPHP3 promoter activity in the cells with the HIF-1α-GFP overexpression was enhanced approximately 1.9 times compared with the control (Figure 3G), which was inhibited by the interference of HIF-1α with the second siHIF-1α (Figure 3H). Protein (Figure 3I left) and transcript (Figure 3I right) level of HIF-1α were decreased by the interference of HIF-1α with the second siHIF-1α in the group with HIF-1α-GFP overexpression. ROS-regulating antioxidant enzymes in the HIF-1α-GFP-overexpressed group were also examined to match the data in Figure 1H. The HIF-1α overexpression enhanced the level in SOD1 ≈1.7 and catalase ≈1.8 times but reduced SOD2 to ≈0.6 and HO-1 to ≈0.5 times (Figure 3J). This suggested that HIF-1α could regulate NPHP3 expression and PC formation under normoxic conditions. 

### 2.3. SD-Induced ROS Are Associated with NPHP3 Expression and HIF-1α-Mediated PC Formation 

We examined whether SD-induced ROS could affect PC formation and NPHP3 expression. As shown in Figure 4A,B, ROS increased in SD conditions, which was inhibited by N-acetylcysteine (NAC) treatment as judged by the changes in mean fluorescence intensity (MFI). The effect of SD-induced ROS on PC formation was confirmed by an NAC treatment [35]. Our data also showed that PC frequency in HeLa cells incubated with SD condition decreased with the NAC treatment ≈30% (Figure 4C). NPHP3 promoter activity in SD condition was also reduced with the NAC treatment ≈70% (Figure 4D). This suggested that ROS could control the SD-induced increase in NPHP3-promoter activity and PC formation. Our data were re-examined using a treatment with H_2_O_2_. The HIF-1α amount was increased by H_2_O_2_ treatment (Figure 4E). ROS-induced PC formation could be confirmed by an increase in PC frequency ≈2.8 times in the H_2_O_2_-treated group (Figure 4F). HIF-1α-mediated PC formation in the H_2_O_2_-treated group was proven by a decrease in PC frequency ≈25% in the second siHIF-1α-treated and H_2_O_2_-treated groups (Figure 4G). This suggested that SD-induced PC formation could be regulated by ROS production and HIF-1α expression.

### 2.4. ROS-Increased ERK Is Related to PC Formation and NPHP3 Transcription 

As ERK is a ROS-induced kinase that regulates HIF-1α in the expression of various genes [37,38], we tested whether ROS-increased ERK was related to PC formation and NPHP3 transcription. As SD increased ERK phosphorylation ≈2.4 times (Figure 5A) that was decreased ≈85% with the NAC treatment (Figure 5B), we confirmed the increase of ROS-mediated ERK phosphorylation ≈3.2 times in the H_2_O_2_-treated group (Figure 5C). We also examined the role of ERK in PC formation and NPHP3 transcription using PD98059 inhibiting ERK activation ≈62% (Figure 5D). PD98059 reduced NPHP3 transcriptional activity ≈75% (Figure 5E) and PC formation ≈21% (Figure 5F). The data demonstrated that ERK could regulate PC formation and NPHP3 expression in SD conditions.

### 2.5. PC Formation Influences NPHP3 Expression and Cell Growth in SD Conditions

We examined whether PC formation could regulate NPHP3 expression and cell growth using ciliobrevin A (CilioA) to inhibit ciliogenesis. PC frequency was inhibited with the CilioA treatment ≈36% (Figure 6A). NPHP3 transcription (Figure 6B) and promoter activity (Figure 6C) decreased with the CilioA treatment in response to incubation under SD conditions. The CilioA treatment reduced cancer cell viability ≈20% (Figure 6D left). The CilioA treatment also reduced the total cell number ≈20% and ≈50% in an incubation with 5% FBS or SD conditions, respectively (Figure 6D right). Decrease in cancer cell viability ≈5% (Figure 6E left) and total cell number ≈20% (Figure 6E right) was also measured in PC-deficient IFT88^−/−^ cells. The data demonstrated that cancer cells under SD conditions were maintained by NPHP3-associated PC formation via ROS-induced HIF-1α and ERK activation (Figure 6F). This suggested that PC formation could control cell viability and growth in an incubation with SD conditions. 

## 3. Discussion

The PC regulates the cell cycle as well as differentiation, polarity, migration and it also maintains homeostasis in tissues and organs [15,39,40] by sensing changes in the enrichment of nutrients [15]. NPHP3, one of the ciliary proteins, contributes to the most frequent genetic disease of renal failure [17,18,19]. PC formation is induced by incubation with a serum-deprivation (SD) medium [11,12,13,14]. ROS are produced as byproducts of cellular respiration under SD conditions [1] and induce an apoptotic response in various cell types [7,8]. In the adaptation of cells to SD-induced oxidative stress, HIF-1α stabilization and HIF-1 activation inhibit apoptosis in cancer cells [23]. However, no information has been reported regarding the role of HIF-1α on NPHP3-associated PC formation to control cell viability under SD normoxic conditions. In this study, our data demonstrated that PC formation regulates cancer cell viability and growth by NPHP3 expression through ROS-induced ERK activation and HIF-1α stabilization under SD conditions. 

Nutrient deprivation primes cancer cells for adaptability to oxidative stress and/or a general survival mechanism to anti-tumorigenic agents [9]. SD changes various cellular conditions [10] including proliferation inhibition [41] and apoptotic cell death [1,36,42,43,44,45,46]. As cancer cells remain alive and grow in SD conditions, it is questionable how cancer cell survival and death are controlled under SD conditions. PC formation is one of the representative SD-induced events in many types of cultured cells [11,12,13,47]. NPHP3, one of the ciliary proteins, regulated PC formation and interacted with thymosin beta-4 to control tumor growth [34,35]. Our data also showed that SD increased PC formation and NPHP3 expression (Figure 1A–F). Defining the molecule working in SD conditions and its mechanism of action on PC formation and NPHP3 expression is required.

The nuclear translocation of the HIF-1α protein is enhanced under normoxic conditions [31] as well as hypoxic conditions [26]. HIF-1α increases renal PC length following renal injury [32]. SD also significantly increase PC length under normoxic conditions [33]. ROS are the representative SD-induced molecules for PC formation [34,35,42]. Our data showed that SD increased the HIF-1α and ROS-regulating proteins (Figure 1G, H) leading to an increase in NPHP3 expression (Figure 2 and Figure 3). Our data also showed that ROS production under SD conditions controlled NPHP3 expression and HIF-1α-mediated PC formation (Figure 4). In contrast, it has been reported that no changes in ROS production and PC formation were detected by the incubation with MitoTempo, an antioxidant for mitochondrial ROS [36]. Oxidative stresses caused by excessive ROS production contribute to the activation of Nrf2 which is a master transcription factor involved in antioxidant and detoxification responses [5,6] to prevent oxidant injury [48]. Further research is required to study whether Nrf2 regulates NPHP3 expression in response to ROS. 

SD-mediated cellular events are regulated by various signaling molecules [49,50,51]. The PC regulates HIF signaling during inflammation [52]. PC formation inhibits cancer cell proliferation [53] or is correlated with a few proliferative cancer cells [54]. ERK is a ROS-induced kinase that regulates HIF-1α for the expression of various genes [37,38]. Our results showed that SD-activated ERK controlled NPHP3 transcription and PC formation (Figure 5), leading to the regulation of tumor cell viability and growth (Figure 6). It is possible that other signaling molecules could be associated with HIF-1α-mediated NPHP3 expression and PC formation. 

The PC assembly responding to SD is associated with the inhibition of Aurora A (AurA) and the inactivation of receptor tyrosine kinases (RTKs) as well as ubiquitin-specific peptidase 8 (USP8) [55,56]. SD may change various intracellular signaling molecules including PI3K/Akt, MEK/ERK1/2, p38 and JNK1/2 [49,51,57]. In terms of cancer cell death, SD has promoted apoptosis by activating the ASK1-JNK/p38 MAPK pathways in hepatocellular carcinomas [58]. SD-induced CD133 prevented colon cancer cell death through the activation of Akt-mediated protein synthesis [59]. SD also promoted tumor cell survival by inducing Bcl 10 through the activation of forkhead transcription factor (FOXO3a)-mediated nuclear factor kappaB [60]. SD also significantly enhanced autophagy-related protein expression [61] and Ca^++^ mobilization from ER [50]. Thus, the relevance of HIF-1α and the signaling molecules involved in PC formation and NPHP3 expression to regulate cancer cell viability are further clarified.

Previous reports showed that the effect of hypoxic conditions on PC formation, length and function is a little complicated with cell type and developmental stage. The percent of elongated cilia reduced significantly in rat tail tendon fascicles under hypoxic conditions, but the cilia length and function increased in the hypoxic fetal proximal and distal collecting epithelia [62,63]. It might be due to many other molecular changes as well as HIF-1α under hypoxic conditions. Among them, hypoxia-induced ROS [27,28,29,30] are common molecules produced by SD-induced oxidative stress [23] in cancer cells. Our data suggest that HIF-1α could be regulated by ROS, leading to PC formation via NPHP3 expression. However, it does not rule out that the Hela cells, which have been cultivated for more than 71 years, will change their functionality. It is also required a controlled experiment to confirm our suggestion in a different cell line and/or a cell line without PC formation such as IFT88^−/−^ cells we used.

Oxygen tension significantly affects the cilium length with the re-localization of GLI2 and SMO proteins into the cilia [33]. Many signaling molecules have been associated with PC membranes, which regulate cell proliferation, migration, and differentiation [56]. Important signaling pathways are mediated by hedgehog (HH), Notch, wigless (Wnt), hippo (Salvador-Warts-Hippo), JAK/STAT, TRPV4, cAMP/cGMP, mTOR and platelet-derived growth factor (PDGF) for intercellular communications between cancer cells [39,40,64,65]. It is, therefore, possible for ROS to stimulate those signaling pathways for NPHP3 expression and PC formation. Whether and how those signaling pathways regulate cancer cell viability and growth under SD conditions remains to be defined. 

Taken together, although many questions remain regarding the relationship between cancer cell survival and the HIF-1α-mediated regulation of NPHP3 expression and PC formation, our data suggest that HIF-1α might regulate NPHP3 expression and PC formation leading to the maintenance of cancer cell survival (Figure 6E). However, unknown molecules induced by SD affecting PC formation and NPHP3 expression cannot be ruled out. Our findings might be re-affirmed by using cells without PC formation. Furthermore, it is also required to study the relevance of the PC formation by the supply of nutrients except SD in the tumor microenvironment using primary samples or 3D cultures. Our results are beneficial to better understand the regulatory molecules that maintain cancer cell survival under SD conditions. 

## 4. Materials and Methods

### 4.1. Reagents

Dimethyloxallyl Glycine (DMOG, 71210) was from Cayman Chemical (Ann Arbor, MI, USA). N-acetyl-L-cysteine (NAC), hydrogen peroxide (H_2_O_2_), MTT [3(4,5-dimethyl-thiazol-2-yl)-2,5-diphenyl tetrazolium bromide], sodium butyrate and 4′,6-diamidno-2-phenylinole (DAPI) were purchased from the Sigma Chemical Co. (St. Louis, MO, USA). 2’,7’-dichlorofluorescin diacetate (DCF-DA) was purchased from Molecular Probe (Eugene, OR, USA). Ciliobrevin A was from Selleck Chemicals (Houston, TX, USA). PD98059 was from Enzo Life Sciences, Inc. (Farmingdale, NY, USA). Mouse antibodies which are reactive with acetylated tubulin (T7451), and β-tubulin (T4026) were from Sigma-Aldrich Co. (St. Louis, MO, USA). Rabbit antibodies which are reactive with Nrf2 and HIF-1α were from Cell signaling Technology (Cat# 12721, Danvers, MA, USA) and Novus Biologicals (Centennial, CO, USA), respectively. Rabbit antibodies which are reactive with Arl13b (17711-1-AP) were from Proteintech Group Inc. (Rosemont, IL, USA). Chicken anti-mouse IgG-Alexa 488 (A-21200) and goat anti-rabbit-Alexa 568 (A-11011) were obtained from Invitrogen (Calsbad, CA, USA). Except where indicated, all other materials are obtained from the Sigma Chemical Co. (St. Louis, MO, USA).

### 4.2. Plasmids and siRNAs

Pre-designed promoters for NPHP3 (NM_153240) was obtained from GeneCopoeia Inc. (Rockville, MD, USA). NPHP3-promoter (HPRM12542) was 1309 bp (−1311 ~ −3) upstream from starting codon for NPHP3 transcription in Homo sapiens 3 BAC RP11-39E4 (AC055732.16) [34]. NPHP3 promoter was cloned at the sites between EcoRI and HindIII of Gaussia luciferase (Gluc) reporter plasmid vector, pEZX-PG02.

Small interference (si) RNAs are customer-ordered to Bioneer (Daejeon, Republic of Korea). Sequences of siRNAs for HIF-1α are as follows: 1st siHIF-1α with [sense: UGA TAC CAA CAG UAA CCA A(dTdT); anti-sense: UUG GUU ACU GUU GGU AUC A(dTdT)] or 2nd siHIF-1α with [sense: CAU GAA AGC ACA GAU GAA U(dTdT); anti-sense: AUU CAU CUG UGC UUU CAU G(dTdT)]. AccuTarget™ negative control siRNA (SN-1001) was also purchased from Bioneer (Daejeon, Republic of Korea).

### 4.3. Cell Culture

HeLa human cervical cancer cells (ATCC # CCL-2) were obtained from Korea Research Institute of Bioscience and Biotechnology (KRIBB) cell bank (Daejeon, Republic of Korea). PC-deficient IFT88^−/−^ A375 human melanoma cells (ATCC # CRL-1619) were kindly provided from Professor Ja-Hyun Koo, Seoul National University (Seoul, Republic of Korea). Cells were cultured as monolayers in Dullecco’s modified Eagle’s medium (DMEM) supplemented with 10% fetal bovine serum (FBS) (GIBCO, Grand Island, NY, USA), 2 mM L-glutamine, 100 units/mL penicillin and streptomycin (GIBCO, Grand Island, NY, USA). Cells were incubated at 37 °C in a humidified atmosphere of 5% CO_2_ maintenance. For the induction of primary cilium formation, cells were incubated in serum-deprived media with 0.1% FBS for 36 h.

### 4.4. Cytotoxicity Assay

Cell survival was measured by using the trypan blue exclusion assay. Confluent cells were cultured with various concentrations of FBS for 24 h. Cells were collected and mixed with equal volume of 0.4% trypan blue in PBS. Dying or dead cells were stained with blue color and viable cells were unstained. Each cell was counted by using hemocytometer under light microscope (Olympus Korea Co., Ltd., Seoul, Republic of Korea).

### 4.5. Hypoxia Treatment

To incubate HeLa cells under hypoxic conditions, cells were placed in an atmosphere of 84% N_2_, 10% H_2_, 5% CO_2_ and 1% O_2_ in a hypoxia chamber (Forma Anaerobic System, Thermo Electron Corporation, Waltham, MA, USA). Cultures were intermittently flushed with nitrogen, sealed, and then maintained in a humidified incubator at 37 °C inside the hypoxia chamber. Hypoxia-treated cells were collected in the hypoxia chamber to avoid the degradation of hypoxia-responsive molecules. 

### 4.6. ROS Measurement 

The level of reactive oxygen species (ROS) was measured by incubating cells with or without 10 μM DCF-DA at 37 ℃ for 30 min. Fluorescence intensity of 10,000 cells was analyzed by FACSCalibur™ (Becton Dickinson, San Joes, CA, USA) [66]. Each cell was acquired with FL1 channel to analyze ROS production by the changes in fluorescence intensity. Cells were gated into one population at dot plot with forward scatter (FSC) in the X-axis and side scatter (SSC) in the Y-axis. Gated cells were represented in the histogram to fluorescence intensity and the geometric mean fluorescence intensity (MFI) was quantitated by WinMDI 2.8 for each sample.

### 4.7. Detection of Primary Cilia

For the detection of primary cilia in vitro, cells were maintained in serum-deprived culture medium [34]. Briefly, HeLa cells were grown on coverslip and then incubated with serum-deprived DMEM with 0.1% FBS for 36 h. Cells were fixed with 4% paraformaldehyde for 10 min, washed three times with cold PBS, and permeabilized with PBST (0.1% (*v*/*v*) Triton X-100 in PBS) for 10 min. Then, cells were washed three times, and incubated with monoclonal anti-acetylated tubulin antibodies diluted (1:1000) or/and polyclonal anti-Arl13b antibodies diluted (1:1000) in PBST for 2 h at room temperature. After washing three times with PBS, cells were incubated with chicken anti-mouse IgG-Alexa 488 diluted (1:1000) or/and goat anti-rabbit IgG-Alexa 568 diluted (1:2000) in PBST for 1h at room temperature. The nucleus was visualized by staining cells with DAPI. After washing with PBS, cells were mounted on slide glass. Primary cilia were observed and photographed at 1000× magnification under a fluorescence microscope (Nikon, Tokyo, Japan).

### 4.8. Transfection of Nucleic Acids

Each plasmid DNA, siRNAs for HIF-1α and AccuTarget™ negative contol siRNA were transfected into cells as follows [34]. Briefly, each nucleic acid and lipofectamine 2000 (Invitrogen, Calsbad, CA, USA) was diluted in serum-free medium and incubated for 5 min, respectively. The diluted nucleic acid and lipofectamine 2000 reagent was mixed by inverting and incubated for 20 min to form complexes. Meanwhile, cells were stabilized by the incubation with the culture medium without antibiotics and serum for at least 2 h prior to the transfection. Pre-formed complexes were added directly to cell culture medium and cells were incubated for an additional 6 h. Then, the culture medium was replaced with antibiotic and 10% FBS-containing DMEM and incubated for 24–72 h prior to each experiment. 

### 4.9. Gaussia Luciferase Assay for Promoter Activity

HeLa cells were transfected with the NPHP3-Gluc plasmids using lipofectamine 2000 (Invitrogen, Carlsbad, CA, USA). Then, cells were incubated for an appropriate time. Secreted Gluc reporter protein was obtained by the collection of culture-conditioned media after the indicated time intervals. Gluc activity of reporter protein was measured by Gluc assay kit (Promega Co., Madison, WI, USA) including coelenterazine as a substrate for Gluc according to the manufacturer’s protocol. Luminescence was detected by using Lumet 3, LB 9508 luminometer (Berthold Technologies GmbH & Co. KG, Bad Wildbad, Germany) [34].

### 4.10. Chromatin Immunoprecipitation (ChIP) Assay

ChIP assays were performed, as described previously [67,68,69]. Cells were crosslinked with a final concentration 1% formaldehyde for 10 min at room temperature. Then, 125 mM of glycine was added to quench unreacted formaldehyde. Cells were collected and sonicated to make DNA fragments with a size range of 200 to 1000 bp. Cell extracts were immune-precipitated using 1 μg anti-HIF-1α or rabbit IgG control (Abcam, Cambridge, UK) for each sample suspended in 450 μL ChIP dilution buffer (0.01% SDS, 1.1% Triton X-100, 1.2 mM EDTA, 16.7 mM Tris-HCl, pH 8.1, 167 mM NaCl) purchased from Cell signaling Technology (Cat# 20-153, Danvers, MA, USA). For all ChIP experiments, PCR analyses were performed by using multiple sets of primers spanning the transcription factor binding site on the NPHP3 gene promoter.

### 4.11. Reverse Transcription Polymerase Chain Reaction (RT-PCR)

Total RNA was extracted by using NucleoZOL reagent (MACHEREY-NAGEL GmbH & Co. KG, Duren, Germany). Complementary DNA (cDNA) was synthesized from 1 μg of isolated total RNA, oligo-dT_18_, and superscript reverse transcriptase (Bioneer, Daejeon, Republic of Korea) in a final volume of 20 μL. For standard PCR, 1 μL of template cDNA was amplified with Taq DNA polymerase. PCR amplification was performed with 30~35 thermocycles for 30 s at 95 °C, 30 s at 55 °C, and 60 s at 72 °C using human oligonucleotide primers specific for NPHP3 (sense: 5′-AGC GAA ATA CCA AGC AAT GG-3′; anti-sense: 5′-TGG AAG GTT CAC TTC CCA AG-3′), HIF-1α (sense: 5′-CTC AAA GTC GGA CAG CCT CA-3′; anti-sense: 5′GAT TGC CCC AGC AGT CTA CA-3′) and actin (sense: 5′-GTC ACC AAC TGG GAC GAC AT-3′; anti-sense: 5′-GCA CAG CCT GGA TAG CAA CG-3′). Amplified PCR products were separated by 1.0–2.0% agarose gel electrophoresis and detected on Ugenius 3^®^ gel documentation system (Syngene, Cambridge, UK).

### 4.12. Real-Time Reverse Transcription (Q-PCR) Analysis

To perform real-time reverse transcription (Q-PCR) analyses, total cellular RNA (5 mg) was reverse transcribed into cDNA using SuperScript II (Invitrogen, Carlsbad CA, USA). Real-time PCR was performed using the CFX96 Touch™ Real-Time PCR Detection System (Bio-Rad laboratories, Hercules, CA, USA). The RT reaction product (100 ng) was amplified with THUNDERBIRD SYBR qPCR Mix (TOYOBO CO., Osaka, Japan) using primers specific for human target genes, SOD1 (sense: 5′-GCA AAG GTG GAA ATG AAG A-3′; anti-sense: 5′-TAG CAG GAT AAC AGA TGA GTT-3′), SOD2 (sense: 5′-CAA GCC TGG TAC ATA CTG A-3′; anti-sense: 5′-TTT GAT GGT TGA CAG ATT CTT T-3′), catalase (sense: 5′-AGG GTG GTG CTC CAA ATT AC-3′; anti-sense: 5′-TTG AAT CTC CGC ACT TCT CC-3′), HO-1 (sense: 5′-CTG CTC TGT AAT GTT GTG-3′; anti-sense: 5′-GTG TGA CCT TCT GTT AGT-3′) and β-Actin (sense: 5′-GCC AGG TCA TCA CCA TTG-3′; anti-sense: 5′-GTT GAA GGT AGT TTC GTG GAT-3′). Samples were heated to 95 °C for 1 min and amplified for 40 cycles followed by a final extension step of 72 °C for 10 min. GAPDH was used as an internal control. Relative quantification of each mRNA was analyzed by the comparative threshold cycle (CT) method [35].

### 4.13. Western Blotting

Cells were lysed in an ice-cold RIPA buffer (Triton X-100)-containing protease inhibitors (2 μg/mL aprotinin, 1 μM pepstatin, 1 μg/mL leupeptin, 1 mM phenylmetylsufonyl fluoride (PMSF), 5 mM sodium fluoride (NaF) and 1 mM sodium orthovanadate (Na_3_VO_4_)). The protein concentration of the sample was measured using SMART^TM^ BCA protein assay kit (Pierce 23228) from iNtRON Biotech. Inc. (Seoul, Republic of Korea). The same amount of heat-denatured protein in the sodium dodecyl sulfate (SDS) sample buffer was separated in the sodium dodecyl sulfate polyacrylamide gel electrophoresis (SDS-PAGE), and then transferred to a nitrocellulose membrane using an electro blotter. An equal amount of loaded sample on the membrane was verified by ponceau S staining. The membrane was incubated with blocking solution (5% non-fat skim milk in Tris-buffered saline with Tween 20 (TBST)), and then followed by incubation with the specific primary antibodies. Horse radish peroxidase (HRP)-conjugated secondary antibody was used for target-specific primary antibody. Target bands were visualized by the reaction with enhanced chemiluminescence (ECL) (Dong in LS, ECL-PS250). Immuno-reactive target bands were visualized by the reaction with enhanced chemiluminescence (ECL-PS250) (Dongin LS, Seoul, Republic of Korea) on X-ray film (Agfa HealthCare, Seoul, Republic of Korea) or by the detection of IRdye with Odyssey CLx Infrared Imaging System (LI-COR Biosciences, Lincoln, NE, USA), respectively [34].

### 4.14. Statistical Analysis

Experimental differences were verified for statistical significance using ANOVA and student’s *t*-test. *p* value of <0.05 and <0.01 was considered to be significant as compared to control.

## Figures and Tables

**Figure 1 ijms-23-14500-f001:**
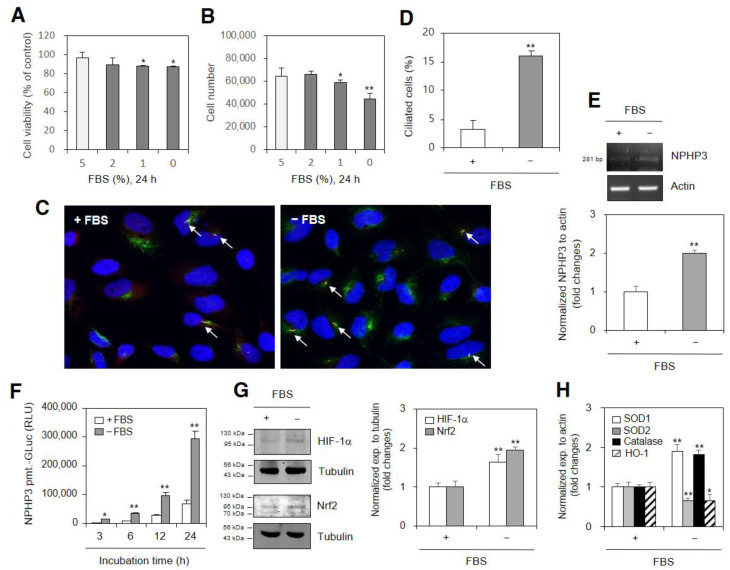
Changes in cell viability and cell number were associated with primary cilium formation, NPHP3 expression and HIF-1α stabilization under serum-deprived (SD) condition. (**A**–**E**) HeLa cells were incubated in a medium with various percentages of fetal bovine serum (FBS) for 24 h. Cell viability was measured by the trypan exclusion assay (**A**). Cell number was counted by using hemocytometer (**B**). The cells were fixed and stained with antibodies against Ac-tubulin (green) and Arl13b (red). Nucleus was stained with DAPI (blue). The primary cilium was observed with 1000× magnification under fluorescence microscope. The image with primary cilia is the representative out of ≈30 pictures. White arrows indicated primary cilia (**C**). The ciliated HeLa cells (*n* > 500 cells) in the presence (white) or absence (grey) of FBS were counted (**D**). Total RNA was prepared by using NucleoZOL^®^ and the expression level of NPHP3 was measured by RT-PCR (**E** top). Density of each band was analyzed by ImageJ 1.34 n and results were normalized to actin (**E** bottom). (**F**) HeLa cells were transfected with pEZX-PG02-NPHP3-promoter Gaussia luciferase (Gluc) plasmids and incubated in the presence or the absence of FBS for up to 24 h. The activity of Gluc in cultured media was measured with luminometer using Gluc substrate. (**G**–**H**) HeLa cells were incubated with or without FBS for 24 h. Cell lysates were prepared. HIF-1α and Nrf2 proteins were detected by Western blot analysis (left). Density of each band was analyzed by ImageJ 1.34 n and results were normalized to tubulin (right). Total RNA was prepared by using NucleoZOL^®^ and the expression level of each antioxidant gene was measured by Q-PCR (**H**). Data were the representative of four experiments. Processing (such as changing brightness and contrast) is applied equally to controls across the entire image (**C**,**E**,**G**). Data in bar graphs represents the means ± SEM. * *p* < 0.05, ** *p* < 0.01; significantly different from 5% FBS-treated control group (**A**,**B**,**D**, **E** bottom, **G** right, **H**) at each time point (**F**).

**Figure 2 ijms-23-14500-f002:**
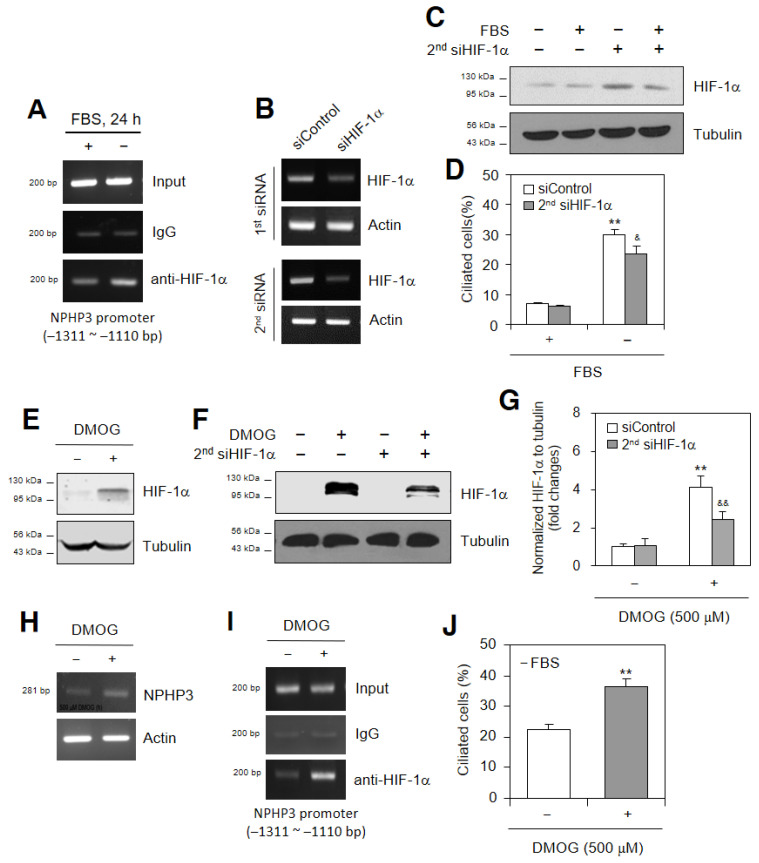
HIF-1α enhanced NPHP3 expression and primary cilium formation under serum-deprived (SD) condition. (**A**) HeLa cells were incubated in the presence or the absence of FBS for 24 h. Cells were fixed with 10% formaldehyde. Their chromatin extracts were immunoprecipitated with anti-HIF-1α antibodies. DNA fragments were subjected to PCR analysis using primer sets spanning the promoter regions. Sequences for primer sets were 5′-CCCTGATCCACATGGAGAATTCA-3′ (forward) and 5′-CTTTAGATTTTCTTCAGGAA-3′ (reverse). Primer set corresponds to −1311 to −1110 bp on NPHP3 promoter. (**B**–**D**) Cells were transfected with AccuTarget™ scrambled siRNA (siControl) or 1st or 2nd HIF-1α-siRNA (siHIF-1α) for 24 h. The cells were incubated in the presence or absence of FBS. The mRNA expression of HIF-1α was detected by RT-PCR (**B**). Cell lysates were prepared and HIF-1α protein was detected by Western blot analysis (**C**). Then, cells were fixed and stained with antibodies against Ac-tubulin and Arl13b. Nucleus was stained with DAPI. The ciliated cells (*n* > 500 cells) in the presence (white) or absence (grey) of FBS were counted (**D**). (**E**–**I**) Cells were treated with 500 μM dimethyloxallyl glycine (DMOG) in the absence (**E**,**H**–**J**) or presence of 2nd siHIF-1α (**F**,**G**). Cell lysates were prepared and HIF-1α protein was detected by Western blot analysis (**E,F**). Density of each band was analyzed by ImageJ 1.34n and results were normalized to tubulin. Relative fold changes in each band to control were represented in bar graph (**G**). Total RNA was prepared by using NucleoZOL^®^ and the expression level of NPHP3 was measured by RT-PCR (**H**). Cells were fixed with 10% formaldehyde. Their chromatin extracts were immunoprecipitated with anti-HIF-1α antibodies. DNA fragments were subjected to PCR analysis using primer sets spanning the promoter regions. Sequences for primer sets were 5′-CCCTGATCCACATGGAGAATTCA-3′ (forward) and 5′-CTTTAGATTTTCTTCAGGAA-3′ (reverse). Primer set corresponds to −1311 to −1110 bp on NPHP3 promoter (**I**). The cells were fixed and stained with antibodies against Ac-tubulin and Arl13b. Nucleus was stained with DAPI. The primary cilium was observed with 1000× magnification under fluorescence microscope. The ciliated HeLa cells (*n* > 500 cells) in the absence (white) or presence (grey) of DMOG were counted (**J**). Data were the representative of four experiments. Processing (such as changing brightness and contrast) is applied equally to controls across the entire image (**A**–**C**,**E**,**F**,**H**,**I**). Data in bar graphs represents the means ± SD. ^&^ *p* < 0.05, ** *p* < 0.01; significantly different from 5% FBS-treated (**D**) or DMOG-untreated (**G**,**J**) control group. ^&&^
*p* < 0.01; significantly different from DMOG-treated and siControl-tansfected group (**G**,**J**).

**Figure 3 ijms-23-14500-f003:**
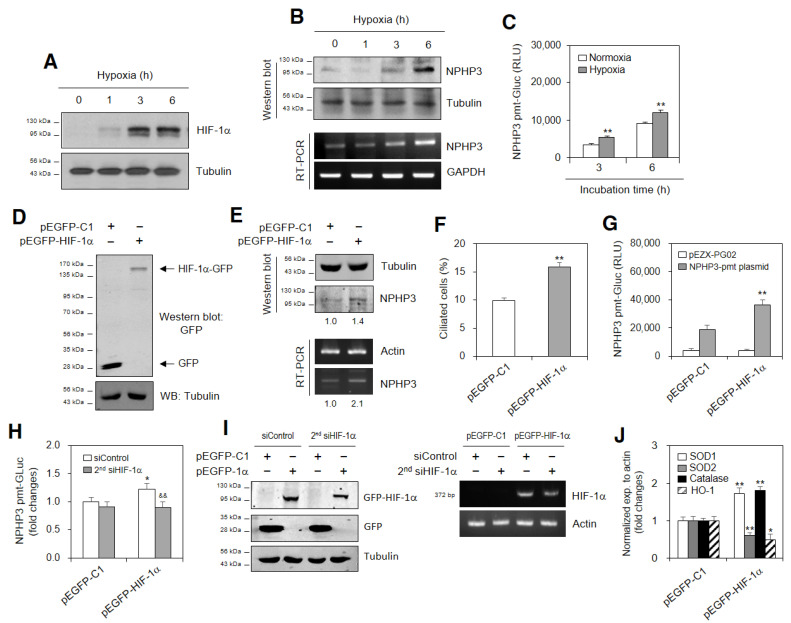
HeLa cells incubated under hypoxic condition increased primary cilium formation. (**A**,**B**) Cells were incubated under normoxic or hypoxic condition for up to 6 h. Cell lysates were prepared. HIF-1α (**A**) and NPHP3 (**B** top) proteins were detected by Western blot analysis. Total RNA was prepared by using NucleoZOL^®^ and the expression level of NPHP3 was measured by RT-PCR (**B** bottom). (**C**) Cells were transfected with pEZX-PG02-NPHP3-promoter Gaussia luciferase (Gluc) plasmid and incubated under normoxic or hypoxic condition for up to 6 h. The activity of Gluc in cultured media was measured with luminometer using Gluc substrate. (**D**–**F**) Cells were transfected with pEGFP-C1 or pEGFP-HIF-1α plasmids and incubated for 24 h. Cell lysatess were prepared and GFP (**D**) and NPHP3 (**E** top) proteins were detected by Western blot analysis. Total RNA was prepared by using NucleoZOL^®^ and the expression level of NPHP3 was measured by RT-PCR (**E** bottom). Density of each band was analyzed by ImageJ 1.34 n and results were normalized to tubulin or actin. Numbers under each band indicated the relative fold changes to control (**E**). The cells were fixed and stained with antibodies against Ac-tubulin and Arl13b. Nucleus was stained with DAPI. The primary cilium was observed with 1000× magnification under fluorescence microscope. The ciliated HeLa cells (*n* > 500 cells) transfected with pEGFP-C1 (white) or pEGFP-HIF-1α (grey) plasmids were counted (**F**). (**G**–**J**) pEZX-PG02 control (white) or pEZX-PG02-NPHP3-promoter-Gluc plasmids (grey) were co-transfected into cells with pEGFP-C1 or pEGFP-HIF-1α plasmids. Cells were incubated for 36 h (**G**). pEZX-PG02-NPHP3-promoter-Gluc plasmids were co-transfected into cells with pEGFP-C1 or pEGFP-HIF-1a plasmids and incubated for 6 h. Then, each group was transfected with AccuTarget™ scrambled siRNA (siControl) or 2nd HIF-1α-siRNA (siHIF-1α) and incubated up to 24 h (**H**). The activity of Gluc in cultured media was measured with luminometer using Gluc substrate (**G**,**H**). Cell lysates were prepared. GFP-HIF-1α and GFP proteins were detected by Western blot analysis (**I** left). Total RNA was prepared by using NucleoZOL^®^ (**I** right, **J**). The expression level of HIF-1α was measured by RT-PCR (**I** right). The expression level of each antioxidant gene was measured by Q-PCR (**J**). Data were the representative of four experiments. Processing (such as changing brightness and contrast) is applied equally to controls across the entire image (**A**,**B**,**D**,**E,I**). Data in bar graphs represents the means ± SD. * *p* < 0.05, ** *p* < 0.01; significantly different from pEGFP-C1-transfected (**F**–**H**,**J**) control group. ^&&^
*p* < 0.01; significantly different from pEGFP-HIF-1α- and siControl-tansfected group (**H**).

**Figure 4 ijms-23-14500-f004:**
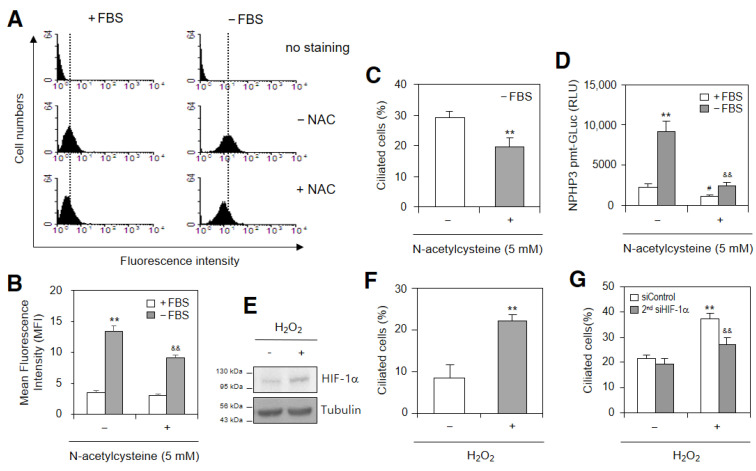
Primary cilium formation was enhanced by reactive oxygen species (ROS) and HIF-1α under serum-deprived (SD) condition. (**A**,**B**) Cells were incubated under SD condition in the absence or presence of N-acetylcysteine (NAC) and treated with DCF-DA. Fluorescent cells with ROS production were measured by FACS analysis (**A**). Geometric mean fluorescence intensity (MFI) from four experiments was analyzed by WinMDI 2.8 (**B**). The differences in MFI were indicated by grey dotted or solid lines. (**C**) Cells were incubated under SD condition in the absence or presence of NAC. The cells were fixed and stained with antibody against Ac-tubulin. The representative image of primary cilia was observed with 400× magnification under fluorescence microscope. The ciliated cells (*n* > 500 cells) in NAC-untreated (white) or -treated (grey) groups were counted. (**D**) Cells were transfected with pEZX-PG02-NPHP3-promoter-Gluc plasmids and incubated in the absence or presence of NAC for 24 h. The activity of Gluc in cultured media was measured with luminometer using Gluc substrate. (**E**) Cells were treated with 50 μM H_2_O_2_ for 24 h. Cell lysates were prepared and HIF-1α protein was detected by Western blot analysis. (**F**,**G**) Cells were treated with 50 μM H_2_O_2_ for 24 h (**F**) after the inhibition of HIF-1α by the transfection of siHIF-1α (**G**). The cells were fixed and stained with antibody against Ac-tubulin. The primary cilium was observed with 1000× magnification under fluorescence microscope. The ciliated cells (*n* > 500 cells) were counted in each group. Data were the representatives of four experiments. Data in bar graphs represent the means ± SD (**B**–**D**,**F**,**G**). ** *p* < 0.01; significantly different from FBS-treated (**B**,**D**) or NAC-untreated (**C**) or H_2_O_2_ –untreated (**F**,**G**) control group. ^#^
*p* < 0.01; significantly different from NAC-untreated group with FBS (**D**). ^&&^
*p* < 0.01; significantly different from SD control and NAC-untreated (**B**,**D**) or H_2_O_2_-treated and control-siRNA-treated group (**G**).

**Figure 5 ijms-23-14500-f005:**
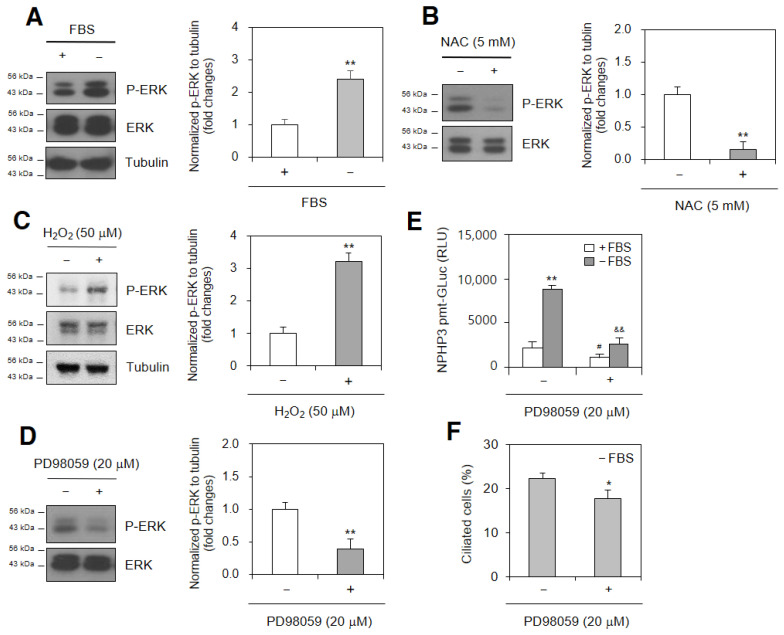
ERK activation under SD condition control primary cilium formation and NPHP3 expression. (**A**–**D**) Cells were incubated with or without 5% FBS in the absence (**A**,**B**) or presence (**B**) of N-acetylcysteine (NAC). Cells were treated with 50 μM H_2_O_2_ for 3 h (**C**). Cells were incubated under serum-deprived (SD) condition in the absence or presence of PD98059, ERK inhibitor (**D**). Cell lysates were prepared and ERK and phospho (p)-ERK proteins were detected by Western blot analysis (**A**–**D** left). Density of each band was analyzed by ImageJ 1.34n and results were normalized to tubulin. Relative fold changes in each band to control were represented in bar graphs (**A**–**D** right). (**E**) Cells were transfected with pEZX-PG02-NPHP3-promoter-Gluc plasmids and incubated with or without 5% FBS in the absence or presence of PD98059 for 24 h. The activity of Gluc in cultured media was measured with luminometer using Gluc substrate. (**F**) Cells were incubated under SD condition in the absence or presence of PD98059. The cells were fixed and stained with antibody against Ac-tubulin. The primary cilium was observed with 1000× magnification under fluorescence microscope. The ciliated cells (*n* > 500 cells) were counted in each group. Data were the representative of four experiments. Processing (such as changing brightness and contrast) is applied equally to controls across the entire image (**A**–**D** left). Data in bar graphs represents the means ± SD (**A**–**D** right, **E**,**F**). * *p* < 0.05, ** *p* < 0.01; significantly different from 5% FBS-treated group (**A**,**B**,**E**) or H_2_O_2_-untreated (**C**) or PD98059-untreated control (**D**,**F**). ^#^*p* < 0.01; significantly different from PD98059-untreated group with FBS. ^&&^
*p* < 0.01; significantly different from PD98059-untreated group without FBS (**E**).

**Figure 6 ijms-23-14500-f006:**
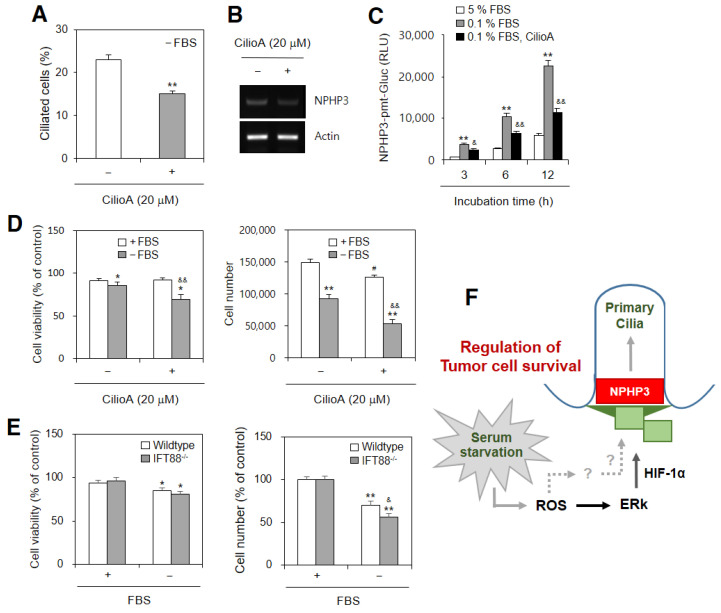
Ciliobrevin A inhibited cell viability, cell number and NPHP3 expression under serum-deprived (SD) condition. (**A**,**B**) Cells were incubated without 5% FBS in the absence or presence of ciliobrevin A (CilioA). Cells were fixed and stained with antibody against Ac-tubulin (green) and DAPI (blue). The primary cilium was observed with 1000× magnification under fluorescence microscope. The ciliated HeLa cells (*n* > 500 cells) in the absence (white) or presence (grey) of CilioA were counted (**A**). Total RNA was prepared by using NucleoZOL^®^ and the expression level of NPHP3 was measured by RT-PCR (**B**). (**C**) Cells were transfected with pEZX-PG02-NPHP3-promoter-Gluc plasmids and incubated in the absence or presence of CilioA for up to 12 h. The activity of Gluc in cultured media was measured with luminometer using Gluc substrate. The activity of Gluc in cultured media was measured with luminometer using Gluc substrate. (**D**,**E**) HeLa (**D**) or PC-deficient IFT88^−/−^ A375 (**E**) cells were incubated with or without FBS for 24 h. Cell viability was measured by the trypan exclusion assay (**D**–**E** left). Cell number was counted using a hemocytometer (**D**,**E** right). Data were representative of four experiments. Processing (such as changing brightness and contrast) is applied equally to controls across the entire image (**B**). Data in bar (**A**,**C**–**E)** graphs represents the means ± SEM. * *p* < 0.05, ** *p* < 0.01; significantly different from CilioA-untreated (**A**) or 5% FBS-treated group (**C**–**E**) at each time point (**C**). ^#^
*p* < 0.01; significantly different from CilioA-untreated group with FBS (**D**). ^&^
*p* < 0.05, ^&&^
*p* < 0.01; significantly different from CilioA-untreated group without FBS (**C**,**D**) at each time point (**C**) or wildtype group without FBS (**E**). (**F**) Scheme for primary cilium formation by serum starvation-induced ROS through ERK activation, HIF-1α and NPHP3 expression to regulate tumor cell survival. While HIF-1α regulated NPHP3 expression and primary cilium formation via ERK activation in HeLa human cervical cancer cells (solid line). ROS also up-regulate NPHP3 expression through unknown another factors (marked as question marks and grey dotted lines). Our findings were indicated by black solid lines. Pathway already known was indicated by grey solid lines.

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
