# Peer review of "HeLa Cervical Cancer Cells Are Maintained by Nephronophthisis 3-Associated Primary Cilium Formation via ROS-Induced ERK and HIF-1α Activation under Serum-Deprived Normoxic Condition"

_ijms, 2022, doi:10.3390/ijms232314500_

Round 1

Reviewer 1 Report (New Reviewer)

This manuscript by Lee et al. describes the role of nephronophthisis 3 (NPHP3) in primary cilium formation in HeLa cervical cancer cells via ROS-induced ERK and HIF-1alpha signaling under serum-deprived condition. Experiments carried out using molecular and pharmacological approaches suggest that serum-deprived toral cellular ROS to regulate NPHP3 expression via ERK. While most of the data support the hypothesis, the link between ROS and HIF-1alpha stabilization ahs not been demonstrated. Further, it is unclear if ERK activation regulates HIF-1alpha in HeLa cells under serum deprivation. Therefore, the suggested signaling mechanism of serum-deprivation ®ROS ®ERK ®HIF-1alph ®enhanced primary cilia formation is incomplete that needs more experimentation to connect the dots. In addition, there are several other major concerns and deficiencies that would need to be addressed.

Major Concerns:

1. Lack of data to show the link between ROS and HIF-1alpha and ERK dependent stabilization of HIF-1alpha in HeLa cells under conditions of serum deprivation.

2. The authors have measured total cellular ROS only and the contribution of mitochondrial ROS in NPHP3 expression and cilia formation should be explored.

3. Fig 2: A – it should be NPHP3 and anti-HIF-1alpha. Data shown in D-H with dimethyloxallyl glycine (DMOG) serves as a control for HIF-1alpha stabilization. It will be critical to show Western blots for serum-deprived conditions (A-C) in addition o RT-PCR data.

4. Fig 3: B, NPHP3 expression at 3 h hypoxic exposure not convincing and clear. Show a better blot and quantify the data. Similarly, Western blot depicting NPHP3 expression in Fig. 3E not convincing.

5. Experiments with PD98059 to inhibit ERK- the concentration of PD98059 (20 µM) is very high and at this pharmacological concentration, there could be non-specific effect on other kinases. Use of PD98059 from 1-10 µM recommended to avoid non-specificity of the kinase inhibitor.

6. Add to the supplement all the original Western blots without cropping. Also, indicate the M.Wt (Kda) of the target protein in the Western blots.

Minor comments:

1. Fig 1G: Should be Nrf2 and not Nrd2

2. Describe hypoxia conditions used for the experiments.

Author Response

Please find the response file attached.

Reviewer 2 Report (New Reviewer)

In this paper, the authors show the relation between primary cilium (PC) growth and survival of cancer cells. To induce PC growth, cells are exposed to serum deprivation (SD) and in consequence, the authors detect ROS induced ERK which regulates HIF1alpha and has an effect on NPHP3 expression and transcriptional activity as well as primary cilium growth and cell survival. The topic is interesting and there are still a lot of unknowns in the field of PC and cancer. However, this paper needs a lot of remodelling before being published. Indeed, the data comes as a juxtaposition of results without explanations of why and sometimes how the experiments are done. The missing links between the different parts, lack of information and conclusions as well as the poor English makes it difficult for the reader to understand the in which order the events happen and if and how they are all connected. The discussion helps a little by telling which ends are still open and need further investigation. However, the result part needs to be much more detailed and explicative. 

Author Response

Please find the response file attached.

Round 2

Reviewer 2 Report (New Reviewer)

Thank you for the changes and clarifications. I understand your situation with the IFT88-/- cells and will wait for you next manuscript on the topic to know more.

This manuscript is a resubmission of an earlier submission. The following is a list of the peer review reports and author responses from that submission.

Round 1

Reviewer 1 Report

The authors present an interesting approach, but the manuscript does not meet the journal's requirements. I suggest revising the entire manuscript, become aware of their question, and catching up on relevant experiments.

A question that arises directly while reading, but could not be answered in the introduction, is the relevance of the PC in the tumor microenvironment. What is exciting, but also not open to discussion, is that cells develop mechanisms in vitro to ensure the supply of nutrients - and of course, they are dependent on FCS. Is it the same in tumors? Primary samples or 3D cultures would bring clarity.

The spectrum of methods is quite large, so it is a pity that these are not used extensively and the data processed/presented. Western blot, PCR, FACS results are shown as representative only - there are no quantifications. Statistical significance is given, but the test is not named. Flow cytometry was used to show the MFI of DCFDA (a quantification is also missing here - and a gating strategy). Alternatively, DCFDA can also be measured by microscopy. Finally, perhaps a hint to give the story more quality overall - a cell line without PC formation or a KO of PC would confirm the hypothesis.

Apart from that, authors should check spelling, grammar, and other typographical errors to read the manuscript fluently; some examples are below:

page 1, line 12: "Primary cilium(PC) is a microtubule-based" instead of "Primary cilium(PC) is microtubule-based";

page 1, line 13-14: “nephronophthisis 3(NPHP3)” - space is missing;

page 1, line 13-14: "interaction with by thymosin β4" – the authors may want to decide for "with" or "by".

… At this point I stopped to read the manuscript and had a look at the figures.

Figure 1: C and D (or descriptions) are swapped; the authors should include the number of biological replicates; >500 cells is a great number for data evaluation (and statistics), but does not correspond to the requirements of biological deviations.

Figure 1, F: Test of statistics should be given

Figure 1, E and G: Is it only one biological replicate? Please repeat and quantify the signal

Figure 4, A: Flow cytometry enables the analysis of thousands of cells and different markers. It would be interesting to see the gating strategy (DCFDA may enter dying cells); Quantification of MFI (with FCS without FCS, with NAC, without NAC) is missing.

Author Response

please refer the response file attached

Reviewer 2 Report

Summary:

Lee et al provides intriguing evidence that cancer cell viability is regulated by NPHP3-mediated primary cilium formation. Primary cilium formation was demonstrated to be associated by serum deprivation. By utilizing serum deprivation, they were able to demonstrate that there was enhanced NPHP3 expression and HIF-1α. siRNA-mediated knockdown of HIF-1α resulted in a reduction ciliated cells. Reactive oxygen species seem to directly affect NPHP3 levels and PC formation, which is impaired by NAC treatment. Erk signaling was demonstrated to play a role in PC formation and NPHP3 expression where inhibition via P98059. However, before the manuscript can be ready for publication, several major and minor revisions must be addressed.

Major revisions

1)    In Figure 1C and 1D, it appears that the figure legends have been swapped. Please place directly on the manuscript, what color represents DAPI, Ac-Tubulin, and, Arl13b. Can the authors provide a better image via confocal microscopy (if they have access to it)? Fluorescence microscopy is not as accurate and the images may prove higher quality.

2)    In Figure 2B, please use 2 unique siRNAs and show the western blot results as well.

3)    In the abstract line 18-19, please adjust the phrasing “The role of HIF-1α on NPHP3 expression was confirmed by siRNA-based 18 inhibition of HIF-1α(siHIF-1α) and the binding of HIF-1α on NPHP3 promoter.” I do not see evidence of siRNA knockdown of HIF-1α on NPHP3 levels (mRNA or protein). Otherwise, provide this evidence please.

4)    Have the authors considered investigating mTOR signaling in relation to PC formation? mTOR has been linked to HIF-1α directly (PMID: 17502379).

5)    The authors should also utilize amino acid deprivation medium to starve cells in addition. They may see a more dramatic effect in PC formation.

6)    The authors demonstrate that in Figure 3F that overexpression of HIF1-α results in enhanced NPHP3 luciferase activity. Please repeat this experiment with siRNA-HIF1-α to show the reciprocal effect.

7)    The authors should try experiments utilizing an actual hypoxia chamber to determine the roles of PC formation and NPHP3, to strongly activate HIF-1α.

Minor revisions

1)    In the materials and methods, there are 2 sections that mention the length of time for serum deprivation, 36 hours or 24-72 hours. Why is there a discrepancy? Please denote the hours use for SD for each experiment.

2)    Figure 2D, please remove “500 uM DMOH (H)” on the image in the main figure of the manuscript.

3)    Line 69-72, there may be a discrepancy in font sizes.

Author Response

please refer the response file attached

Round 2

Reviewer 1 Report

Unfortunately, the manuscript has to be rejected again because it is still insufficient, and suggested improvements have not been incorporated.

Author Response

Please refer the file attached.

Reviewer 2 Report

Thank you for your thoughtful responses and modifications to your story, according to the revisions I suggested.

Author Response

We thank reviewer for giving us a valuable comment.

We have revised English language and style in the revised manuscript.

Round 3

Reviewer 1 Report

I understand that the remaining data from the larger 2021 study will be released. Collecting data costs a lot of money, time, and effort. Nevertheless, this manuscript needs meaningful data before publishing in a journal with an impact factor >5. I recommend revising the manuscript and including missing experiments. Without this revision, this manuscript may be accepted in an appropriate journal (impact <4).